# Determinants of Herders' Satisfaction with the Grassland Ecosystem Compensation Policy: A Case Study of Gansu Province, China

**Sanqiang Du [1], Yunxiang Cheng [2,\*] and Dong An [3]**

1 The United Graduate School of Agricultural Sciences, Tottori University, Tottori 680-8553, Japan; dsanqiang@gmail.com
2 College of Ecology and Environment, Inner Mongolia University, Hohhot 010021, China
3 College of Pastoral Agriculture Science and Technology, Lanzhou University, Lanzhou 730020, China; and20@lzu.edu.cn
\* Correspondence: chengyx@imu.edu.cn

**Abstract:** This study investigated herders' satisfaction with the implementation effects of the Grassland Ecosystem Compensation Policy (GECP) in Sunan (subsidy hierarchization) and Gannan (subsidy harmonization), China. Survey data from 140 randomly selected herder households were analyzed using descriptive statistics and ordered logistic regression to identify the factors influencing herders' satisfaction. The results showed that in Sunan, 47.89% of respondents expressed satisfaction with the GECP. Their satisfaction positively correlated with changes in native grass, ecological compensation income, and reduced inedible grass. Conversely, it exhibited negative associations with the ethnic background of the household head, livestock numbers, and willingness to relocate. In Gannan, a substantial level of dissatisfaction prevailed (69.57%). However, satisfied herders had connections with changes in native grass, income diversity, and ecological compensation income. Significantly, this study highlights that ecological compensation income and changes in native grass consistently influence herders' satisfaction regardless of the subsidy design. These findings offer valuable insights for improving herders' satisfaction with the implementation effects of the GECP in regions with diverse ecological subsidy designs. Additionally, it presents a fresh perspective for scholars to analyze the GECP under different ecological subsidy frameworks further.

**Keywords:** Grassland Ecosystem Compensation Policy; subsidy design; policy satisfaction; determinants; China

## 1. Introduction

Grassland, a crucial component of terrestrial natural resources, is vital in sustaining the natural ecosystem and supporting society and economic development. However, the escalating demand for natural resources driven by economic growth, over-exploitation, and climate change has resulted in grassland degradation, desertification, and soil erosion [1–3]. The deterioration of the grassland ecosystem poses a significant challenge to the sustainable utilization of grassland resources [4,5]. Existing data reveals that approximately 20% of the world's natural ecosystems and 73% of grassland ecosystems have suffered varying degrees of degradation [6]. China, one of the countries rich in grassland resources, boasts a total grassland area of 400 million hectares, accounting for 40% of its land area [7]. Nevertheless, over the past four decades, 90% of the natural grassland has experienced considerable degradation, primarily due to overgrazing [8–10], with 34% classified as severely degraded [11]. Such degradation of the grassland ecosystem has profound consequences for herders who heavily rely on grassland resources for their livelihoods and animal husbandry development [12].

The Chinese government has implemented several national-level ecological protection programs to address the challenges overgrazing poses and its impact on the grassland

ecological environment and regional economic development. These programs include the Ecological Migration Program, Returning Grazing to Grassland Program, the Beijing-Tianjin Sandstorm Source Controlling Program, and the Returning Farmland to Forests Program [13]. In June 2011, the government introduced the "combination of production and ecology, ecological priority" approach in key livestock production areas. This led to the establishment of the Grassland Ecosystem Subsidy and Award Scheme, commonly known as the Grassland Ecosystem Compensation Policy (GECP) [14,15]. The primary objective of the GECP is to safeguard grassland ecology, ensure a stable supply of livestock products, and enhance herder income [16–19]. The government allocates an annual budget of CNY 15 billion to implement the GECP in eight provinces. Subsequently, Hebei and Shanxi provinces (regions) were included in the policy's scope in 2012. This national policy categorizes grasslands into grazing ban areas and grass–livestock balance areas, determined by the ecological conditions of the grasslands. It considers grassland ecology and herders' livelihoods by providing subsidies and awards to grassland contractors who adhere to the policy. This initiative has ushered in new opportunities for economic development in the pasture regions [20]. The GECP has emerged as the most extensive ecological protection program, encompassing numerous provinces, and significantly impacting China's conservation efforts.

Since the initiation of the GECP in 2011, there have been significant improvements in the ecological compensation standard, with cumulative investments exceeding USD 22.39 million. These investments are crucial in enabling herders to benefit from "ecological welfare". China's comprehensive vegetation coverage in its grasslands has increased from 51% in 2011 to 56.1% in 2020 [21]. Moreover, the mode of production has gradually shifted from traditional grazing to half-grazing, half-house feeding, and house-feeding practices. Additionally, continuous improvements in the infrastructure of pastoral areas have been noted [22–26]. To achieve a balance between grassland and livestock, most herders have adjusted their stocking rates by regulating livestock numbers, grassland management areas, and forage purchases. However, despite a consensus among herders regarding grassland ecological improvement, challenges persist, including overall livestock overload and severe overload in certain localities, indicating that complete mitigation of grassland degradation has not been achieved [27,28]. Moreover, herders' effectiveness in implementing the GECP is affected by the unreasonable setting of ecological compensation fund standards [6], issues related to delayed disbursements, and even instances of delinquency [29]. Another concern is the income disparity among herders, where relatively wealthier herders receive more eco-compensation funds than economically disadvantaged ones, potentially widening the income gap [6,30]. In light of these challenges, stakeholders have reached a consensus on the need to enhance the long-term mechanism of the GECP [31,32].

Satisfaction, a crucial form of psychological perception and judgment, can reflect the quality of participants' subjective well-being and their access to the utility value of resources [33]. Public policies that gain participants' recognition and achieve high satisfaction are more likely to perform effectively [34]. For herders, who are direct stakeholders in GECP implementation, the difference between their interest demands and the GECP's interest provisions is manifested in their satisfaction with the policy [35]. Evaluating herders' satisfaction with GECP can objectively reflect the overall impact of the policy's implementation on various aspects of herders' production and life. Moreover, herders' satisfaction directly influences their policy implementation behavior, consequently affecting its overall effectiveness [36].

Various regions experience disparities in subsidies due to differences in natural conditions, socio-economic environments, and levels of economic development [37]. In response to these variations, some local governments have maintained the uniform subsidy standards established by the national government, while others have further refined these standards based on the unique grassland ecological conditions in their regions. This adaptation has led to two main types of subsidies in the current implementation of the GECP: subsidy hierarchization, tailored to different grassland ecological conditions, and subsidy

harmonization, adhering to the unified national standard. Recognizing the significant impact of subsidy design on participants' satisfaction levels within the GECP, which, in turn, influences their behavior in policy adherence and, ultimately, the policy's effectiveness [38], policymakers must consider the factors that shape herders' satisfaction across diverse subsidy design regions. This consideration is pivotal for developing specific and targeted follow-up measures to enhance herders' satisfaction. Additionally, it contributes to the ongoing enrichment of research in GECP satisfaction and its influencing factors. Therefore, this study endeavors to analyze herders' satisfaction with the implementation outcomes of the GECP across various subsidy designs and to identify the determinants thereof.

## 2. Literature Review

The research landscape concerning the Grassland Ecosystem Compensation Policy (GECP) is extensive and multifaceted. Many quantitative analysis studies have compared grassland ecology, livestock production, and herders' income before and after GECP implementation in the same region. These investigations have consistently demonstrated that the GECP yields positive outcomes in protecting grassland ecology, regulating livestock production, and safeguarding herders' livelihoods [39]. However, it is notable that the primary obstacle to effective policy implementation continues to be the relatively lower subsidy standards [6,40]. Some studies have delved into formulating grassland ecological subsidy standards using quantitative and qualitative analyses. These studies have underscored the necessity for ecological subsidy standards to consider not only the direct costs but also opportunity costs and the value of grassland ecological service functions within local herders' households [41,42]. Moreover, research focused on herders' satisfaction with the implementation of the GECP and the factors influencing it, as assessed using quantitative analysis, has revealed substantial regional variations in herders' satisfaction with the implementation outcomes of the GECP. These variations are significantly influenced by the social, economic, and ecological characteristics specific to each region [43–45]. Furthermore, studies have compared grassland eco-compensation methods and their effectiveness across countries such as China and Germany. These analyses have revealed that eco-compensation policies in both countries have faced challenges in delivering environmental services due to insufficient participation, monitoring, control, and regulatory provisions [46,47]. Additionally, research has scrutinized herders' satisfaction in the Western Desert Region and the Tibetan Plateau Region in conjunction with the effectiveness of the GECP. These studies have unveiled differences in overall satisfaction, with herders in the Western Desert Region displaying higher satisfaction levels due to diversified income sources and reduced reliance on animal husbandry. The size of households and herders' comprehension of the policy have been identified as crucial determinants of herders' policy satisfaction in all regions [48,49]. Although extensive research has been conducted on satisfaction and its influencing factors regarding implementing the GECP, all these studies are based on regions with the same subsidy standards and various ecological functional zones. Nonetheless, there is still a notable gap in exploring herders' satisfaction and the factors influencing it in regions with diverse subsidy standard designs.

## 3. Materials and Methods

### 3.1. Study Area

Sunan Yugu Ethnic Minority Autonomous County (Sunan) is centrally located in Gansu Province, spanning the geographical coordinates of 97°20′ to 102°12′ E longitude and 37°28′ to 39°04′ N latitude (Figure 1). This county encompasses a total land area of approximately $2.38 \times 10^6$ ha, with grassland constituting the dominant land resource. The region's elevation varies from 1327 to 5564 m above sea level, and it receives an annual average of 2200 to 3100 h of sunshine. Temperature fluctuations throughout the year range from 8.0 to 16.7 °C. Annual precipitation levels range from 66 to 600 mm, and the frost-free period extends from 50 to 140 days [50]. As of 2018, the county had a population of 39,500,

with ethnic minorities constituting 56.5%. It encompassed 15.57 ha of various grassland types and supported a livestock population of 1,388,800 composed of various types, with a slaughter rate of over 50%. Animal husbandry contributed to approximately 65% or more of the county's income of farmers and herders, establishing it as a significant region for the cattle and sheep industry in Gansu Province [51,52].

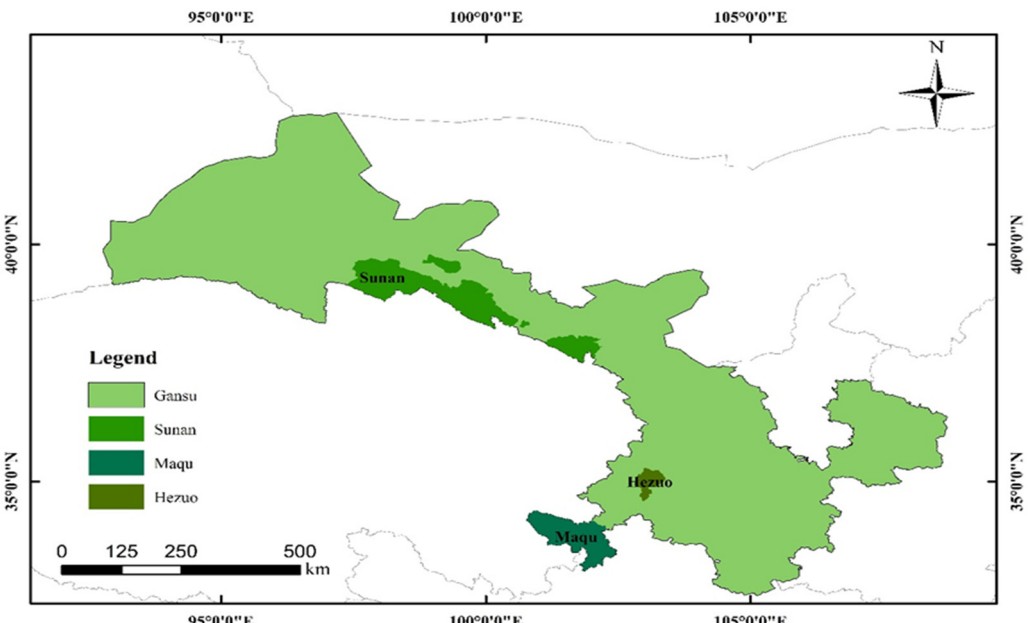

**Figure 1.** Field survey areas in Gansu Province, China.

Gannan Tibetan Autonomous Prefecture (Gannan) is located on the northeastern edge of the Tibetan Plateau, situated within the geographical coordinates of 100°46′ to 104°44′ E longitude and 33°06′ to 36°10′ N latitude. It has a total land area spanning $4.5 \times 10^6$ ha and serves as a crucial water conservation and replenishment region for the Yellow and Yangtze Rivers. The elevation in Gannan varies from 1100 to 4900 m above sea level, with annual sunshine hours ranging between 1800 and 2600 h. The region experiences annual temperature fluctuations from −3 to 13 °C, an average annual precipitation of 505 mm, and an annual frost-free period lasting 120 days. As of 2019, the population of Gannan was approximately 730,700, with Tibetans constituting 54.2% of the total population. The region encompasses $32.73 \times 10^4$ ha of natural grassland, accounting for 70.28% of the total land area. Within this expanse are $15.57 \times 10^4$ ha of usable grassland, with 80% of the natural grasslands concentrated in patches. This concentration makes it the natural grassland on the Tibetan Plateau with the highest stocking rate and most significant resistance to grazing, boasting a theoretical stocking rate of 6.2 million sheep units [53,54].

*3.2. Overview of the Grassland Ecosystem Compensation Policy*

The Grassland Ecosystem Compensation Policy (GECP) comprises two principal components: the grazing ban and the grass–animal balance policies. Areas falling under the grazing ban are designated as off-limits for livestock production, while regions outside the ban, suitable for livestock grazing, are categorized as grass–animal balance areas. This policy operates on a five-year cycle, with evaluations conducted at the end of each cycle to determine whether to maintain the grazing ban or transition to grass–animal balance areas. This decision is contingent upon the restoration of grassland ecological function [55].

The implementation of the GECP involves three specific measures: (1) For severely degraded grasslands, the grazing ban is enforced, with compensation rates set at USD 47.80 per hectare in the Qinghai–Tibet Plateau (QTP) area, USD 10.19 per hectare in the Loess Plateau area, and USD 8.54 per hectare in the Western Desert area. (2) Grasslands located outside the grazing ban area are subject to stocking rate verification to ensure a balanced

grass–animal relationship. Compensation rates for these areas are established at USD 7.39 per hectare in the QTP area, USD 5.89 per hectare in the Loess Plateau area, and USD 4.79 per hectare in the Western Desert area. (3) The Ministry of Finance and the Ministry of Agriculture conduct inspections and supervision to assess the effectiveness of ecological compensation implementation in different regions. Regions demonstrating noteworthy results receive financial rewards [56].

Under the national policy, Sunan has further divided the ecological compensation standard into five levels, considering grassland quality, yield, and stocking rate, and has implemented subsidy hierarchization measures. In contrast, Gannan adheres to the national policy without further subdivision, following the subsidy harmonization approach (Table 1).

**Table 1.** Implementation measures of GESAS.

| Region | Item | Measure |
|---|---|---|
| Gannan [a] | Grazing bans subsidy | USD 47.80 [c]/ha/year |
| | Grass–animal balance award | USD 7.39/ha/year |
| | Productive comprehensive subsidy | USD 588.24/household/year |
| | Complementary measure | Resource census, ecological monitoring, grassland management |
| Sunan [b] | Grazing bans subsidy | first level = USD 176.89/ha<br>second level = USD 108.35/ha<br>third level = USD 39.82/ha<br>fourth level = USD 24.64/ha<br>fifth level = USD 13.57/ha |
| | Grass–animal balance award | first level = USD 84.75/ha<br>second level = USD 49.37/ha<br>third level = USD 19.81/ha<br>fourth level = USD 13.26/ha<br>fifth level = USD 4.46/ha |
| | Productive comprehensive subsidy | USD 441.18/household/year |
| | Protecting minimum subsidy, limiting maximum subsidy | Herders with a pasture area of ≤35 ha are protected by a minimum subsidy (USD 493.09/person); the total amount of grazing ban subsidy does not exceed USD 5626.47/person; the total amount of subsidy for partial grazing ban and partial grass–animal balance subsidy does not exceed USD 4155.88/person. |

Note: [a] Gannan Tibetan Autonomous Prefecture implements a new round of the Grassland Ecological Protection Subsidy and Award Scheme Implementation Plan (2016–2020). [b] Sunan Yugu Autonomous County implements a new round of the Grassland Ecological Protection Subsidy and Award Scheme Implementation Plan (2016–2020). [c] Based on the exchange rate of the People's Bank of China in May 2017, USD 1 = CNY 6.8. Source: authors' calculations from field study data.

### 3.3. Data Collection

In this study, two rounds of combined telephone and face-to-face questionnaire surveys were conducted with farmer and herder households in the study area during the summers of 2017 and 2018. To refine the questionnaire, a pilot survey was conducted before the first formal survey. Respondents who had already been interviewed were excluded from the second round of surveys to avoid duplicating data. Notably, most interviewees were Tibetan or Mongolian individuals who did not speak Mandarin. To mitigate potential data bias arising from language differences, the research team enlisted the assistance of two local undergraduates who could understand Tibetan to conduct face-to-face interviews and assist with questionnaire completion. All enumerators and interpreters underwent uniform training before conducting the formal survey to ensure consistency and avoid potential response bias. The questionnaire included basic questions covering: (1) demographic

characteristics of participants (e.g., gender, age, education level, etc.); (2) family economy (e.g., household business income, policy transfer income, etc.); environmental perception (e.g., changes in native grass, changes in inedible native grass, etc.); and (3) perception of living conditions (e.g., willingness to move into urban areas, etc.).

To ensure the authenticity and reliability of the research data, the following principles were adhered to during the field research: (1) enumerators explained the survey's purpose and obtained voluntary consent from participants before administering each questionnaire; (2) preference was given to interviewees who were more familiar with the family situation; and (3) to enable the herders to express their personal views truthfully and to avoid strategic behavior, village cadres were not involved in the field research. In addition, all enumerators also held discussions with the Grassland Management Office to gain a comprehensive understanding of the implementation of the GECP.

Due to the extensive size and low population density of the survey area, this study selected Huangcheng, Baiyin, and Dahe, which have a large implementation area in Sunan, as well as Maqu and Hezuo in Gannan, as the sample areas based on the actual grassland area involved in the GECP. Finally, 142 households from 11 townships and 35 villages were surveyed for this study using a random sampling method. Two households were excluded from the analysis due to data quality issues, resulting in 140 households with valid data. Among these, 71 households were from Sunan and 69 were from Gannan, with a questionnaire validity rate of 98.59%.

### 3.4. Variables Selection and Measurement

In this study, the main focus was on the satisfaction of herders with the implementation effect of the GECP, which we refer to as "policy satisfaction". To ensure rigorous measurement of the explained variable and accurate data collection, this study used the Likert 5-Point Scale. This scale allowed for the precise representation of quantitative data about practical issues. Drawing from the research of [57,58], the researchers measured policy satisfaction by asking participants to rate their overall satisfaction with the implementation effects of the GECP. The scale ranged from "very dissatisfied" to "not very satisfied", "general", "relatively satisfied", and "very satisfied", which were assigned values of 1, 2, 3, 4, and 5, respectively. This method evaluated participants' satisfaction levels with implementing the policy.

In the model used to analyze policy satisfaction factors, this study considered various independent variables based on the literature on the GECP in northern China [12,29,31,49] and the socio-economic characteristics of farmers and herders in the study area. These independent variables included individual basic characteristics, family characteristics, cognitive characteristics, and regional characteristics of herders. The descriptive statistics of these variables are presented in Table 2.

**Table 2.** Description of the characteristics of the sample households.

| Variable | Def. | Sunan (n = 71) | | Gannan (n = 69) | | Overall (N = 140) | |
|---|---|---|---|---|---|---|---|
| | | Mean | S.D. | Mean | S.D. | Mean | S.D. |
| Satisfaction with GECP [a] | 1−5 [b] | 3.42 | 0.89 | 2.26 | 0.68 | 2.85 | 0.98 |
| Demographic and economic characteristics | | | | | | | |
| Age ($X_1$) | year | 42.37 | 11.46 | 42.33 | 14.98 | 42.35 | 13.27 |
| Ethnic group ($X_2$) | 0−1 [c] | 0.20 | 0.40 | 0.00 | 0.00 | 0.10 | 0.30 |
| Educational level ($X_3$) | 0−9 [d] | 3.80 | 1.98 | 1.30 | 1.61 | 2.57 | 2.19 |
| Household size ($X_4$) | person | 3.30 | 0.70 | 4.55 | 1.79 | 3.91 | 1.49 |
| Pasture area ($X_5$) | Ha [e] | 294.25 | 425.24 | 52.81 | 81.50 | 175.26 | 330.14 |
| Number of livestock ($X_6$) | SSU [f] | 983.03 | 751.40 | 676.30 | 1186.15 | 831.86 | 998.16 |
| Leased-in pasture ($X_7$) | 0−1 [g] | 0.52 | 0.50 | 0.07 | 0.26 | 0.30 | 0.46 |
| Diversity of income sources ($X_8$) | quantity | 2.68 | 0.67 | 2.25 | 0.69 | 2.46 | 0.71 |
| Ecological compensation income ($X_9$) | USD [h] | 3487.55 | 4504.19 | 1463.11 | 1226.90 | 2489.79 | 3461.94 |

**Table 2.** *Cont.*

| Variable | Def. | Sunan (n = 71) | | Gannan (n = 69) | | Overall (N = 140) | |
|---|---|---|---|---|---|---|---|
| | | Mean | S.D. | Mean | S.D. | Mean | S.D. |
| Ecological and environmental characteristics | | | | | | | |
| Changes in native grass ($X_{10}$) | $1-5$ [b] | 3.45 | 0.81 | 2.40 | 0.77 | 2.94 | 0.95 |
| Changes in inedible native grass ($X_{11}$) | $0-2$ [i] | 0.23 | 0.64 | 1.25 | 0.74 | 0.73 | 0.86 |
| Life situation characteristics | | | | | | | |
| Willingness to move to urban area ($X_{12}$) | $0-1$ [g] | 0.32 | 0.47 | 0.43 | 0.50 | 0.38 | 0.49 |

Note: Def. is an abbreviation of definition. [a] GECP = Grassland Ecosystem Compensation Policy. [b] 1 = very dissatisfied (decrease a lot), 2 = dissatisfied (decrease), 3 = neither satisfied nor unsatisfied (no change), 4 = satisfied (increase), 5 = very satisfied (increase a lot). [c] 0 = Han, 1 = Tibetan or Mongolian. [d] 0 = illiterate, 1 = literate, never been to school, 2 = primary school, 3 = middle school, 4 = high school, 5 = undergraduate, 6 = technical secondary school, 7 = junior college, 8 = above a bachelor's degree, 9 = others. [e] 1 ha = 15 mu. [f] SSU is the abbreviation of standard sheep unit, 1 cow = 5 SSUs, 1 horse = 6 SSUs. [g] 0 = no, 1 = yes. [h] Based on the exchange rate of the People's Bank of China in May 2017, USD 1 = CNY 6.8. [i] 0 = no, 1 = yes, 2 = not sure. Source: authors' calculations from field study data.

### 3.4.1. Demographic and Economic Characteristics

Age ($X_1$): The age of the household head can exert either positive or negative influences on satisfaction with the implementation effects of the GECP. In broad terms, the labor capacity of the household workforce tends to diminish as the household head ages, and older herders may exhibit a greater inclination to accept ecological compensation funds instead of continued manual labor [59]. Conversely, younger herders are often more open to exploring new opportunities and transitioning into different occupations, while older herders may face the risk of livelihood disruption if they choose to depart from animal husbandry, potentially impacting their overall satisfaction with the GECP [60].

Ethnic group ($X_2$): The ethnicity of household heads could potentially have a negative impact on satisfaction with the implementation of the GECP. Ethnic minorities often rely heavily on grassland animal husbandry as their primary source of livelihood, and they have a strong cultural and economic attachment to grassland resources. The enforcement of the GECP may require ethnic minority herders to leave the grasslands and abandon their traditional animal husbandry practices, which can lead to dissatisfaction and challenges in adapting to alternative livelihoods [61].

Educational level ($X_3$): The educational level of the household head can have either positive or negative implications for satisfaction with the implementation of the GECP. Highly educated herders often exhibit a heightened awareness of national policies, enabling them to recognize the long-term significance of GECP implementation and potential benefits [61]. Conversely, the educational level of the household head can serve as an indicator of their production capacity, enhancing their ability to manage animal husbandry operations and effectively mitigate risks, which can, in turn, make them more inclined to expand their production scale [62].

Household size ($X_4$): Household size can have either a positive or negative impact on satisfaction with the implementation of the GECP. A larger household size has the potential to release more labor for other forms of work after GECP implementation, contributing to increased household income for herders [61]. Conversely, grassland animal husbandry is a labor-intensive industry, and a larger household size means more available labor resources, potentially leading to a greater willingness among herders to expand their business operations [59].

Pasture area ($X_5$): Pasture area can have a positive impact on satisfaction with the implementation of the GECP. This is because the amount of ecological compensation is typically determined by the size of the pasture area, meaning that herder households with larger pasture areas are likely to receive more substantial ecological compensation funds [59].

Number of livestock ($X_6$): The number of livestock can have either positive or negative effects on satisfaction with the implementation effects of the GECP. Herder households with more livestock can generate substantial income from animal husbandry and receive significant funds for ecological compensation [63]. However, as the income from the sale of livestock is typically the primary source of income for herder households, a reduction in the number of livestock, as required by the GECP, often leads to a decrease in household income [13]. This reduction in income can negatively influence satisfaction levels.

Leased-in pasture ($X_7$): The use of leased-in pasture can have either positive or negative effects on satisfaction with the implementation effects of the GECP. Although implementing the GECP resulted in some small-scale household businesses abandoning animal husbandry, large-scale business households realized economies of scale by leasing pastures [64]. On the other hand, while herder households can increase their livestock population through leased-in pastures, the ecological compensation funds for these leased pastures typically go to the owners of the pastures rather than to the users [65].

Diversity of income sources ($X_8$): Having a diverse range of income sources may have a positive impact on satisfaction with the implementation effects of the GECP. A diversified income portfolio can enhance the economic stability of herder households, helping to offset any adverse effects on animal husbandry incomes resulting from the implementation of the GECP [66].

Ecological compensation income ($X_9$): Ecological compensation income can have either positive or negative effects on satisfaction with the implementation effects of the GECP. Ecological compensation funds can help offset losses in animal husbandry income, potentially increasing satisfaction. On the other hand, higher ecological compensation income may indicate that herder households are reducing their livestock numbers, which could negatively impact their overall income [63].

### 3.4.2. Ecological and Environmental Characteristics

Changes in native grass ($X_{10}$): Changes in native grasses can have either positive or negative effects on satisfaction with the implementation effects of the GECP. On the one hand, a lower degree of grassland degradation and better-quality native grasses can improve livestock quality, potentially leading to higher satisfaction [63]. On the other hand, herder households might prioritize immediate material benefits from animal husbandry over the uncertain long-term prospects of grassland ecosystem restoration [65].

Changes in inedible native grass ($X_{11}$): Changes in inedible native grasses can have a negative effect on satisfaction with the implementation effects of the GECP. An increase in the proportion of inedible native grasses signifies ongoing deterioration of the grassland ecosystem, which, in turn, places a greater financial burden on herder households to purchase commercial forage [67].

### 3.4.3. Life Situation Characteristics

Willingness to move to an urban area ($X_{12}$): Willingness to relocate to urban areas may have a negative impact on satisfaction with the implementation effects of the GECP. Given that livestock rearing constitutes the primary livelihood of herder households, implementing the GECP can compel unskilled herders to seek new livelihoods in urban areas, thereby increasing the burden on herder households [49].

### *3.5. Economic Modeling*

### 3.5.1. Ordinal Logistic Regression Analysis

In this study, the appropriate analysis model is an ordered logistic model, as the dependent variable (i.e., herders' satisfaction with the implementation effect of the GECP) is an ordered categorical variable with different ranks and degrees. The ordered logistic

model is a discrete choice for analyzing data with ordered categories. The model design is based on previous research [43,68] and is as follows:

$$y_{ij} = \begin{cases} 1 \text{ if customer i belongs to category j} \\ \qquad 0 \text{ otherwise} \end{cases} \tag{1}$$

$$i = 1, 2, \ldots, n, j = 1, 2, \ldots, m$$

$$y_i^* = \alpha' x_i + \mu_i$$

$$\mu_i \sim \text{Logistic} \ (\theta = 1) \tag{2}$$

In the ordered logistic model, the latent variable $y^*$ is related to the independent variable x through the regression coefficient $\alpha$ and the random error term $\mu$. The probability density function of the logistic distribution with a mean value of 0 is:

$$f(x) = \frac{1}{\theta} \frac{\exp\left(\frac{x}{\theta}\right)}{\left(1 + \exp\left(\frac{x}{\theta}\right)\right)^2} \tag{3}$$

Since the latent variable $y^*$ is not directly observed, it is measured with observable values, which are divided into m categories, as shown in Equation (4):

$$y_{i,1} = 1 \text{ if } y_i^* \leq \gamma_1$$

$$y_{i,1} = 1 \text{ if } \gamma_{j-1} < y_i^* \leq \gamma_j \text{ for } j = 2, 3, \cdots, m-1 \tag{4}$$

$$y_{i,m} = 1 \text{ if } \gamma_{m-1} < y_i^*$$

In the equation, the threshold values $\gamma_i$ need to satisfy the condition $\gamma_1 < \gamma_2 < \gamma_3 \cdots < \gamma_{m-1}$. When $\gamma_0 = -\infty$ and $\gamma_m = +\infty$, if $\gamma_{j-1} < y_i^* \leq \gamma_j, j = 1, 2, \cdots, m$, then i belongs to the j class. Combined with Equations (1)–(3):

$$P(\text{customer i belongs to category j}) = P\left(y_{ij} = 1\right)$$

$$\begin{aligned} = P\left(\gamma_{j-1} < y_i^* \leq \gamma_j\right) &= P\left(\gamma_{j-1} < a'x_i + \mu_i \leq \gamma_j\right) \\ &= P\left(\gamma_{j-1} - a'x_i < \mu_i \leq \gamma_j - a'x_i\right) \\ &= F\left(\gamma_j - a'x_i\right) - F\left(\gamma_{j-1} - a'x_i\right) \end{aligned} \tag{5}$$

In the equation, F represents the cumulative density function of the logistic distribution. Model (5) is an ordered logistic model, and the maximum likelihood method is used for parameter estimation.

### 3.5.2. Multicollinearity Test

Multicollinearity is a phenomenon in linear regression models characterized by high correlations among independent variables, which can result in unstable and imprecise parameter estimates [69]. When two or more predictor variables are strongly correlated, it becomes challenging to discern the individual impact of each variable on the dependent variable. Researchers often turn to the variance inflation factor (VIF) to detect multicollinearity. The VIF quantifies how much the variance in an estimated regression coefficient is inflated due to multicollinearity [70]. A VIF exceeding 10 typically indicates problematic multicollinearity, signifying that the variance in the coefficient estimate is significantly amplified because of the high intercorrelations among variables [71]. In Gannan, mul-

ticollinearity analysis revealed a substantial correlation between the variables "pasture area" and "ecological compensation income" among herder households in pure pastoral areas, resulting in a VIF value of 10.88. Consequently, the variable "pasture area" was omitted from the analysis to mitigate the multicollinearity issue. This exclusion aids in attaining more stable and dependable parameter estimates within the ordered logistic model. Addressing multicollinearity is crucial for upholding the validity and precision of statistical analyses and the accurate interpretation of findings.

### 3.6. Data Management and Analysis Methods

At the end of each day's survey, all questionnaires were thoroughly reviewed to ensure data integrity and accuracy. Upon completion of the entire survey, we meticulously transcribed the data from the returned questionnaires into an electronic Microsoft Excel dataset. To maintain data accuracy, we performed regular comparisons between the Excel dataset and the original questionnaires, enabling us to identify and rectify any potential data entry errors that might have occurred during the transcription process.

The data were analyzed using STATA 16 and SPSS 20. Firstly, we conducted a descriptive analysis using SPSS 20 to report household demographics and livestock assets. Secondly, as the model is based on cross-sectional data, the possibility of correlations between two or more variables leading to multicollinearity was addressed. A comprehensive analysis was performed using STATA 16 on the relevant variables to mitigate the impact of multicollinearity before proceeding with the multiple-ordered logistic regression analysis. Finally, the multiple-ordered logistic regression model was applied using STATA 16 to the cross-sectional data of 140 households to analyze the determinants of herders' satisfaction with the implementation effects of GECP.

## 4. Results and Discussion

### 4.1. Herders' Satisfaction with the Implementation Effect of the GECP

Figure 2 illustrates the distribution of herders' satisfaction levels regarding the implementation effect of the Grassland Ecological Compensation Policy (GECP) in Sunan (with subsidy hierarchization) and Gannan (with subsidy harmonization). In the overall assessment, the predominant level of satisfaction was 'dissatisfaction', with 44.29% of respondents expressing this sentiment. In contrast, 27.86% of herder households reported 'satisfaction', and a minor fraction, 3.57%, indicated they were 'very satisfied'. More specifically, in Sunan, 47.89% of herder households reported being 'satisfied' with implementing the GECP. Additionally, 25.35% of respondents fell into the 'uncertain' category, meaning they were neither satisfied nor dissatisfied. Only a smaller fraction, 7.04%, expressed being 'very satisfied'. On the other hand, in Gannan, a significant majority, 69.57% of herder households, conveyed 'dissatisfaction' with the implementation effect of the GECP. This was followed by 17.39% of herders who reported being 'uncertain' about their satisfaction levels. A notably smaller proportion, 7.25%, expressed 'satisfaction' with the implementation effect of the GECP.

Economic benefits were identified as the most influential factor affecting participant satisfaction during the GECP implementation process [6]. The varying satisfaction levels between the Sunan and Gannan regions can be attributed, in part, to differences in ecological compensation income resulting from distinct subsidy designs. An examination of the ecological subsidy standards in Sunan and Gannan (Table 1) reveals that the grazing ban subsidy of USD 47.80/ha/year in Gannan falls between the second (USD 108.35/ha/year) and third (USD 39.82/ha/year) levels of the same subsidy type in Sunan, leaning closer to the third level. This situation not only fails to provide positive incentives for herder households that maintain better grassland ecological conditions, such as those in the first (USD 176.89/ha) and second levels of grassland ecological conditions, but may also reduce their motivation to adhere to the GECP strictly. Consequently, this can lead to the gradual deterioration of grassland ecological conditions. Furthermore, it might cause herder households in regions with poorer grassland ecological conditions, such as those

in the fourth (USD 24.64/ha/year) and fifth levels (USD 13.57/ha/year) of grassland ecological conditions, to maintain the current suboptimal conditions while receiving higher ecological subsidies. This can hinder improvements in grassland ecological conditions. Regarding the grass–animal balance subsidy, the subsidy standard of USD 7.39/ha/year in Gannan is approximately between the fourth (USD 13.26/ha/year) and fifth levels (USD 4.46/ha/year) of the same subsidy type in Sunan, leaning closer to the fifth level. Such a design significantly affects the enthusiasm and motivation of herder households whose grassland ecological status can reach the fourth level and beyond to adhere strictly to the GECP. Over time, this may result in the deterioration of grassland ecological conditions to a less favorable state. While the productive comprehensive subsidy in Gannan is higher than that of Sunan by USD 147.06/ha/year, it remains uniform for herder households across different grassland ecological conditions. Consequently, it does not create disparities in ecological compensation income or incentives for GECP implementation among herder households.

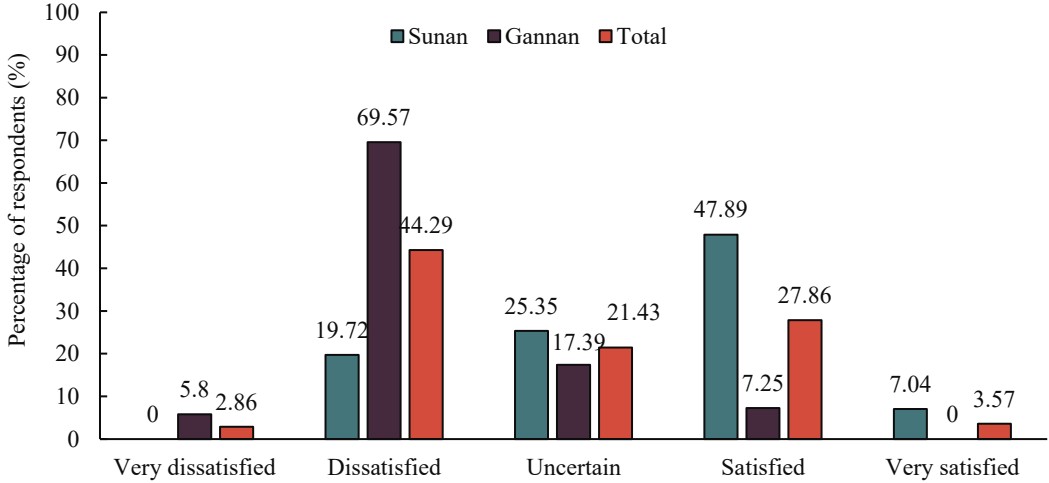

**Figure 2.** Distribution of policy satisfaction among sample households. Note: Uncertain = neither satisfied nor dissatisfied.

Based on the field survey conducted in this study (Appendix A Table A1), it was observed that in Sunan, 53.52% of herder households received ecological compensation income within the range of USD 2500 to USD 5000, followed by 22.54%, with compensation falling between USD 5000 and USD 7500. In contrast, in Gannan, 84.06% of herder households received ecological compensation income within the lower range of USD 0 to USD 2500, with only 11.59% receiving compensation between USD 2500 and USD 4500. These disparities in compensation income distribution likely contributed to the differing satisfaction levels with the GECP implementation effect between the two regions.

*4.2. Determinants of Herders' Satisfaction with the GECP*

4.2.1. Influencing Factors on Herders' Satisfaction with the GECP in Sunan (with Subsidy Hierarchization)

Table 3 illustrates the factors influencing herders' satisfaction with the implementation effect of the GECP in Sunan. These factors include changes in native grasses, the ethnic background of the household head, the number of livestock reared, willingness to move to an urban area, ecological compensation income, and changes in inedible native grasses. Notably, changes in native grasses showed a significant positive correlation with policy satisfaction at the 1% significance level. Within the context of grassland livestock husbandry, native grasses play a pivotal role. Apart from the winter season, when herders are compelled to purchase commercial forage, livestock development heavily relies on the abundant native grasses within pastures during the remaining seasons. An increase in native grasses signifies an improvement in the ecological conditions of the grasslands.

This enhancement fulfills the forage requirements for livestock growth and enables herder households to accumulate hay reserves, reducing their dependency on expensive commercial forage during the winter. Consequently, this substantially contributes to their overall satisfaction with the policy. Conversely, the ethnic background of the household head, the number of livestock reared, and the willingness to move to urban areas exhibited significant negative correlations with policy satisfaction. In Sunan, livestock breeders are predominantly Tibetans or Mongolians, while Han people primarily cultivate crops with animal husbandry as a secondary occupation. Implementing the GECP has resulted in a significant reduction in the number of livestock maintained by herder households, with this impact being more pronounced among ethnic minorities than among the Han population. Consequently, ethnic minorities express reduced satisfaction with the outcomes of GECP implementation. Traditionally, the total livestock count is a measure of wealth among herders, reflecting an individual's economic status [72,73]. The strict constraints imposed by the GECP on the allowable number of livestock per unit of grassland area have led to a decrease in the total livestock holdings of herder households, particularly those with larger cattle herds. This reduction has contributed to their decreased satisfaction with the outcomes of GECP implementation. Furthermore, some herders find themselves compelled to seek job opportunities in urban areas due to their inability to rely solely on animal husbandry to support their livelihoods. This shift results from reduced or restricted livestock holdings and decreased ecological subsidies. Many herders possess minimal skills beyond livestock rearing and often have limited formal education. This predicament further diminishes their satisfaction with the execution of the GECP. These findings align with results reported in [63,74,75] based on field surveys conducted in northwestern China.

**Table 3.** Regression results of the ordered logistic model.

| Variable | Sunan | | | Gannan | | |
|---|---|---|---|---|---|---|
| | Coef. | Std. Err. | z | Coef. | Std. Err. | z |
| Age | −0.02 | 0.04 | −0.37 | −0.01 | 0.02 | −0.31 |
| Ethnic group | −3.00 *** | 0.96 | −3.14 | — | — | — |
| Household size | 0.04 | 0.49 | 0.09 | −0.05 | 0.18 | −0.26 |
| Educational level | −0.13 | 0.22 | −0.57 | −0.01 | 0.26 | −0.05 |
| Pasture area | −0.00 | 0.00 | −0.22 | — | — | — |
| Number of livestock | −0.00 *** | 0.00 | −2.99 | −0.00 | 0.00 | −0.57 |
| Leased-in pasture | 1.27 | 0.90 | 1.41 | −0.34 | 1.14 | −0.03 |
| Diversity of income sources | −0.08 | 0.49 | −0.17 | 1.03 ** | 0.50 | 2.08 |
| Ecological compensation income | 0.00 * | 0.00 | 1.89 | 0.00 * | 0.00 | 1.84 |
| Changes in native grass | 1.56 *** | 0.48 | 3.21 | 1.97 *** | 0.48 | 4.07 |
| Changes in inedible native grass | 0.94 * | 0.52 | 1.82 | 0.21 | 0.51 | 0.42 |
| Willingness to move to urban areas | −2.98 *** | 0.91 | −3.27 | 0.47 | 0.81 | 0.58 |
| No. of observations | 71 | | | 69 | | |
| Log likelihood | −64.793 | | | −46.211 | | |
| *Prob* > chi$^2$ | 0.000 | | | 0.000 | | |
| Pseudo R$^2$ | 0.244 | | | 0.266 | | |

Note: ***, **, and * indicate significance at 1%, 5%, and 10% respectively. Coef. and Std. Err. are abbreviations for coefficient and standard error, respectively. Source: authors' estimations from field study data.

Ecological compensation income and changes in inedible native grasses exhibit a significant positive correlation with policy satisfaction at the 10% significance level. The primary objective of herders in livestock production is to secure economic income. While the GECP mandates a reduction in the number of livestock reared by herder households, resulting in decreased income from animal husbandry, the provision of ecological compensation income helps alleviate the economic losses experienced by these households. Moreover, the proliferation of inedible native grasses in the grassland ecosystem competes with edible native grasses for essential nutrients and living space. As the prevalence of inedible native grasses decreases, livestock can consume more edible native grasses, pro-

moting their growth and overall well-being. These findings support conclusions drawn from previous studies [61,73,75]. These studies have emphasized the crucial role of ecological compensation income and the grassland ecological environment in shaping herders' satisfaction with the GECP.

4.2.2. Influencing Factors on Herders' Satisfaction with the GECP in Gannan (with Subsidy Harmonization)

Table 3 shows significant correlations between herders' satisfaction with the policy and several influential factors in Gannan. Notably, changes in native grasses exhibit a strong positive correlation with policy satisfaction, reaching the 1% significance level. Similarly, the diversity of income sources and ecological compensation income display significant positive correlations at the 5% and 10% significance levels, respectively. The presence of healthy and sufficient native grasses is fundamental to the success of grassland animal husbandry. An increase in native grasses ensures an abundant supply of fodder for livestock during foraging and reduces the reliance of herder households on commercial fodder, resulting in cost savings. As a result, herders' satisfaction with the policy is positively impacted. Furthermore, diversifying income sources is crucial in reducing the dependency of herder households on animal husbandry. This diversification plays a crucial role in mitigating the adverse effects of a substantial reduction in livestock income on the overall living standards of herder households. Additionally, the provision of ecological compensation income significantly contributes to enhancing herders' economic well-being. Although the amount may be limited, it serves as a valuable safety net against a significant decline in their quality of life resulting from reduced numbers of livestock reared. These findings align with and further reinforce the evidence presented in previous studies [75,76].

**5. Conclusions**

Based on data collected from 140 herder households in Sunan (with subsidy hierarchization) and Gannan (with subsidy harmonization), two different ecological subsidy design regions in Gansu Province, China, this study used descriptive statistical analyses and ordered logistic regression modeling to investigate herders' satisfaction with the implementation effects of the Grassland Ecosystem Compensation Policy and identify its determinants. The results showed variations in herders' satisfaction based on the different ecological subsidy designs. In Sunan, 47.89% of respondents expressed satisfaction with the policy. Their satisfaction was positively associated with factors such as changes in native grasses on pastures, ecological compensation income, and alterations in inedible native grasses on pastures. On the other hand, it was negatively correlated with the ethnic background of the household head, the number of livestock reared, and the willingness to relocate to an urban area. In contrast, 69.57% of respondents expressed dissatisfaction with the policy in Gannan. Factors positively associated with satisfaction in this region included changes in native grasses on pastures, diversity of income sources, and ecological compensation income. Interestingly, both ecological compensation income and changes in native grasses exhibited a significant positive correlation with herders' satisfaction, regardless of the specific subsidy design in place. These findings provide valuable insights into the factors influencing herders' satisfaction with the Grassland Ecosystem Compensation Policy. They highlight the importance of ecological compensation income and the condition of native grasses in shaping herders' perceptions of the policy's effectiveness. This study's results agree with previous research, which indicated that herders often evaluate the Grassland Ecosystem Compensation Policy as moderate, satisfied, or dissatisfied, with fewer expressing very high satisfaction [4,6,50]. Additionally, the subsidy amount and improved grassland ecological conditions were critical concerns for herders [6,76]. This study enriches the understanding of herders' satisfaction with the policy and provides a new perspective for scholars in the same field to analyze Grassland Ecosystem Compensation Policy implementation in different regions.

Although the Grassland Ecosystem Compensation Policy is currently in its third round of implementation (2022–2026), it is essential to note that the study area in this research still follows the subsidy design used during the second round of the Grassland Ecosystem Compensation Policy (2017–2021). Consequently, the findings of this study continue to offer valuable insights into improving herders' satisfaction with the policy's implementation effects across different subsidy design regions. In the context of Sunan, policymakers are urged to consider several strategic avenues:

(1) Policymakers should strongly emphasize ethnic diversity and ensure the protection of the interests of ethnic minorities within the policy framework. However, additional support for ethnic groups may raise concerns about fairness among Han herders. Therefore, it is crucial to maintain a balance in the benefits received by these two types of herders.

(2) It is essential to monitor changes in inedible native grasses closely and implement measures to prevent their unchecked proliferation. Long-term monitoring, guided by the expertise of herders and ecologists who can identify inedible native grass species, is necessary.

(3) Policymakers should provide clear and well-defined policy guidance to reshape herders' traditional perceptions of wealth, which often revolve around livestock numbers. However, reshaping herders' perceptions of wealth is a long-term process that requires careful consideration of herders' acceptance and understanding.

(4) To address the propensity of some herders to migrate to urban areas, policymakers should establish re-employment support programs, including skills training and identifying organizations that can absorb this workforce.

In the context of Gannan, policymakers are encouraged to optimize the diversification of herders' household income sources. However, it is crucial to consider new income sources that align with local socio-economic characteristics and the capacities of herders.

Regardless of the specific subsidy design region, this study emphasizes the importance of making timely adjustments to ecological compensation funds. These adjustments should be context-based, considering the different grassland ecological conditions and local socio-economic development levels. Simultaneously, this study underscores the necessity of preserving and enhancing the grassland ecological environment in all regions where these policies are implemented. By actively embracing and implementing these recommendations, policymakers can make significant strides toward enhancing the satisfaction levels of herder households with grassland ecosystem compensation policies.

This study primarily focuses on herders' overall satisfaction and the factors influencing the Grassland Ecosystem Compensation Policy. However, it acknowledges certain limitations, particularly regarding the specific analysis of herders' satisfaction with the grazing ban and the grass−livestock balance policies. These limitations stem from herders' uncertainty about the specific types of ecological subsidies they received and their corresponding grassland areas during the field survey. Furthermore, due to constraints related to time and funding, this study did not explore the long-term and dynamic feedback of herders on the implementation effects of the Grassland Ecosystem Compensation Policy. These limitations allow future researchers to conduct separate, long-term analyses of grazing ban subsidies and grass–animal balance subsidies in various ecological subsidy design regions.

**Author Contributions:** Conceptualization, S.D. and Y.C.; methodology, S.D. and Y.C.; software, S.D. and D.A.; validation, Y.C. and D.A.; formal analysis, S.D. and D.A.; investigation, S.D.; resources, Y.C.; data curation, S.D. and Y.C.; writing—original draft preparation, S.D.; writing—review and editing, S.D., Y.C. and D.A.; visualization, S.D., Y.C. and D.A.; supervision, Y.C.; project administration, Y.C.; funding acquisition, Y.C. All authors have read and agreed to the published version of the manuscript.

**Funding:** This research was funded by the National Natural Science Foundation of Inner Mongolia (No. 2021MS03032), Research Foundation for Advanced Talents of Inner Mongolia University (10000-21311201/018).

**Institutional Review Board Statement:** Not applicable.

**Informed Consent Statement:** Informed consent was obtained from all subjects involved in this study.

**Data Availability Statement:** The data sources used for analysis are presented in the text.

**Acknowledgments:** The authors sincerely thank Kumi Yasunobu and Asres Elias of Tottori University, Japan, for their invaluable guidance in developing this article. Additionally, the authors sincerely appreciate the assistance Choje Nauri, a Tibetan student, provided during the field survey.

**Conflicts of Interest:** The authors declare no conflict of interest.

## Appendix A

**Table A1.** Distribution of ecological compensation income of herder households.

| Compensation Income (USD) | Sunan (n = 71) | | Gannan (n = 69) | | Overall (N = 140) | |
|---|---|---|---|---|---|---|
| | Freq | % | Freq | % | Freq | % |
| [0, 2500) | 3 | 4.23 | 58 | 84.06 | 61 | 43.57 |
| [2500, 5000) | 38 | 53.52 | 8 | 11.59 | 46 | 32.86 |
| [5000, 7500) | 16 | 22.54 | 2 | 2.90 | 18 | 12.86 |
| [7500, 10,000) | 0 | 0 | 0 | 0 | 0 | 0 |
| [10,000, 12,500) | 0 | 0 | 1 | 1.45 | 1 | 0.71 |
| [12,500, 15,000) | 2 | 2.82 | 0 | 0 | 2 | 1.43 |
| [15,000, 17,500) | 8 | 11.27 | 0 | 0 | 8 | 5.71 |
| [17,500, 20,000) | 0 | 0 | 0 | 0 | 0 | 0 |
| [20,000, 22,500) | 2 | 2.82 | 0 | 0 | 2 | 1.43 |
| [22,500, 25,000) | 2 | 2.82 | 0 | 0 | 2 | 1.43 |
| Total | 71 | 100 | 69 | 100 | 140 | 100 |
| Min | 2352.94 | | 588.24 | | 588.24 | |
| Max | 23,382.35 | | 10,441.18 | | 23,382.35 | |
| Mean | 6683.96 * (5657.32) | | 1799.22 * (1419.55) | | 4276.48 (4807.33) | |

Note: * denotes the significance level of 0.05 for the difference in means of the corresponding indicator between herder households in each subsidy design region and the overall study households. Freq is the abbreviation for frequency. Source: authors' estimations from field study data.

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
