# Peer review of "Determinants of Herders’ Satisfaction with the Grassland Ecosystem Compensation Policy: A Case Study of Gansu Province, China"

_sustainability, doi:10.3390/su152216123_

Round 1
Reviewer 1 Report
Comments and Suggestions for Authors
The manuscript presents a comprehensive study on the satisfaction levels of herders with the implementation of the Grassland Ecosystem Compensation Policy (GECP) in two regions, Sunan and Gannan, located in Gansu Province, China. The research employs a combination of descriptive statistical analyses and ordered logistic regression modeling to explore the determinants of herders' satisfaction. The study offers valuable insights into how different subsidy designs affect herder satisfaction and provides policy recommendations based on the findings. While, I recommend this paper until the authors address some major problems. The detailed comments please see the attachment.

Reviewer 2 Report
Comments and Suggestions for Authors
Authors investigate the herders satisfaction with the implementation effects of the grassland ecosystem subsidy and award in two different regions of China. However, authors need to address the following issues before it can be accepted for publication.
General comments: Authors need to improve the English language especially grammar and sentence making to further polish the paper. Some statements are very long which must be shorten for the clarity of general readers.
Specific comments:
Line 61: How conflict will be solved between herder's and land owner regarding subsidy distribution?
Line 110: What mechanism was adapted for subsidy distribution?
Line 111: How this study is unique from the other published studies with the same aspects? What is the innovation?
Line 169: Why there was a huge difference between subsidies of two regions? what is the justification?
Line 186-87: Which statistical approach was applied to determine the number of questionnaire and sample size? Which method or formula was used?
Line 261: Authors just have talked about the leased pasture here. Classify the tenure system of grassland which respect to the study area.
Line 359: Authors did not compare the finding of this study with previous literature. Only a few citations were used in whole discussion section. Also, there is lack of justification and alignment with current findings. Authors need to work seriously and MUST improve this section by adding ample supportive studies.
Line 449: Conclusions must be concise and be focused on the aspect of present study, instead of replication of abstract.
Comments on the Quality of English LanguageAuthors need to improve the English language especially grammar and sentence making to further polish the paper. Some statements are very long which must be shorten for the clarity of general readers.
Reviewer 3 Report
Comments and Suggestions for Authors
I like this issue, but it is necessary to make more detailed discussions to enhance the depth and content of your research article, here are some additional methods and suggestions to enhance the depth and content of your research article:
1. Comparative Analysis:
Consider conducting a comparative analysis between the two subsidy designs (hierarchization vs. harmonization) beyond just their impact on herders' satisfaction. You could explore how these designs affect other factors, such as ecological outcomes, economic stability, and social cohesion within these regions. This could provide a broader perspective on the consequences of different subsidy approaches.
2. Qualitative Interviews:
Supplement your quantitative data with qualitative interviews or focus group discussions with herders. Qualitative data can provide valuable insights into the reasons behind satisfaction or dissatisfaction and can help you better understand the nuances of their experiences.
3. Spatial Analysis:
If you have geographic data, consider conducting a spatial analysis to examine how the geographic distribution of subsidies corresponds to changes in grassland conditions. This can provide insights into whether the spatial allocation of subsidies is effective in achieving ecological goals.
4. Longitudinal Study:
Expand your research to include a longitudinal study. Tracking changes in herders' satisfaction and grassland conditions over time can reveal trends and help establish causal relationships between subsidies and outcomes.
5. Stakeholder Analysis:
Investigate the perspectives of various stakeholders involved in the subsidy schemes, such as government agencies, environmental organizations, and local communities. Understanding their views and interests can provide a more comprehensive understanding of the policy landscape.
6. Cultural Factors:
Explore the role of cultural factors in herders' satisfaction. Cultural values and traditions often influence people's perceptions and behaviors, so understanding these aspects can add depth to your analysis.
7. Environmental Impact Assessment:
Extend your research to assess the environmental impact of the subsidy schemes. Analyze how changes in grasslands affect local ecosystems, including wildlife, plant biodiversity, and water resources.
8. Cost-Benefit Analysis:
Conduct a cost-benefit analysis of the subsidy schemes. Evaluate whether the economic benefits of the schemes, such as increased income for herders, outweigh the costs, including administrative expenses and potential environmental trade-offs.
9. Policy Analysis:
Undertake a policy analysis to assess the strengths and weaknesses of the grassland ecosystem subsidy and award scheme in its current form. Consider whether there are opportunities for policy improvement.
10. International Comparisons:
Compare the grassland subsidy scheme in Gansu Province with similar schemes in other regions or countries. This international perspective can offer insights into global best practices and potential lessons that can be applied locally.
Other suggestions are:
1. Title:
The title you've provided gives a clear idea of the topic and scope of your study. However, you might consider making it a bit more concise and specific. Here's a revised title:
"Determinants of Herders' Satisfaction with Grassland Ecosystem Subsidy Schemes: A Case Study in Gansu Province, China"
2. Abstract:
Your abstract gives a good overview of the study's objectives and findings. However, to improve clarity and conciseness, consider the following suggestions:
- Sentence Structure: Some sentences in the abstract are quite long and complex. Break them down into shorter, more digestible sentences for easier readability.
- Keywords: Include a list of keywords at the end of the abstract. These should be relevant terms that potential readers might use to search for your article.
3. Introduction:
In the introduction section, provide background information about the grassland ecosystem subsidy and award scheme, its importance, and the specific context in Gansu Province. You may also briefly explain the differences between subsidy designs in Sunan and Gannan, setting the stage for the reader.
4. Methodology:
- Sampling: Provide more details about the sampling method used to select the 140 herder households. Explain why this sample size was chosen and if there were any biases that needed to be addressed.
- Data Collection: Explain the data collection process, including the survey questions used to assess herders' satisfaction.
- Statistical Analysis: Describe the statistical methods used, such as the ordered logistic regression. This will help readers understand how you arrived at your conclusions.
5. Results:
- Consistency in Terminology: Ensure that the terminology used in this section is consistent with the abstract. For example, if you mention "satisfaction" in the abstract, use the same term consistently throughout the article.
6. Discussion:
- Interpretation of Results: Provide a detailed interpretation of the results. Explain why ecological compensation income and changes in native grass consistently drive herders' satisfaction, regardless of subsidy design.
- Policy Implications: Expand on the policy implications mentioned in the abstract. Discuss why strategies to protect ethnic diversity, monitor inedible native grass proliferation, redefine traditional wealth perceptions, and establish re-employment support programs are important in Sunan. Similarly, explain the emphasis on diversifying income sources in Gannan.
7. Conclusion:
- Summary: Summarize the key findings of your study and their broader implications for grassland ecosystem compensation policies in various regions.
- Practical Actionable Recommendations: Provide more practical and actionable recommendations for policymakers and stakeholders based on your findings. How can these recommendations be implemented effectively?
- Areas for Further Study: Suggest areas for future research that can build upon your findings. Are there any unanswered questions or new research directions that your study highlights?
8. References:
- Ensure that all sources cited in the article are included in the references section, and vice versa. Follow a consistent citation style throughout.
Comments on the Quality of English Language
The overall quality of the English language in your research article is good. However, there are a few suggestions to enhance clarity and readability:
- Sentence Structure and Length: Ensure that your sentences are clear and concise. Some sentences in your text are quite lengthy, which can make it challenging for readers to follow the argument. Break long sentences into shorter ones for better readability.
- Punctuation: Pay attention to punctuation, especially the use of commas and periods. Correct placement of commas can make a significant difference in sentence clarity.
- Transition Words: Use transition words and phrases to guide readers through your text. These words help in showing the logical flow of ideas and relationships between sentences and paragraphs.
- Parallel Structure: Maintain parallel structure when listing items or presenting information. This creates symmetry and makes the text more organized. For instance, "ecological compensation income, and changes in inedible grass" can be made parallel by rephrasing.
- Clarity and Consistency: Ensure that terminology and phrases are consistent throughout the paper. For example, you mentioned "grassland ecosystem subsidy and award scheme" and "grassland ecosystem compensation policy." Consistency in terminology will improve clarity.
- Academic Style: The tone and style of your writing are appropriate for an academic paper. However, consider avoiding contractions (e.g., "it's" should be "it is") for formal academic writing.
- Proofreading: Carefully proofread your document for grammatical and typographical errors. Small errors can detract from the overall professionalism of the paper.
- Citations: Double-check your citations to ensure they follow the correct format and style guide (e.g., APA, MLA) consistently throughout the paper.
- Paragraph Structure: Make sure each paragraph focuses on a single idea or topic. This improves clarity and readability.
- Headings and Subheadings: If applicable, consider using headings and subheadings to organize your paper's content. They provide a clear structure for readers to follow.
- Abstract: Ensure that the abstract provides a concise summary of the research objectives, methods, and key findings.
- Data Presentation: When presenting data or statistics, consider using tables, figures, or charts where appropriate. Visual aids can enhance understanding.
Reviewer 4 Report
Comments and Suggestions for Authors
The title should be amended to:
The Impact of Grassland Ecosystem Subsidy and Award Scheme on Herders' Satisfaction: The Case of Gansu Province in China
Abstract needs more info on the objectives and significance of study. Findings of Abstract was repeated and prolonged unnecessarily at the end. Please revise it.
The Introduction missed main concern such as effects of delays in subsidy payments or awards could also lead to dissatisfaction.
The Introduction lacks a solid justification between and how subsidy and award scheme leads to economic growth and subsequently satisfaction of herders.
Introduction and literature review should be updated by recent 3 years relevant research studies.
On top of page 3, What was your study contribution whereas there was many similar research done?
Info on both subsidies of hierarchization and harmonization were not provided in the introduction.
Explain either subsidy hierarchization or subsidy harmonization resulted in Herders’ satisfaction and how do you differentiate and measure it?
Your study area was filled with Geographical positioning info rather than relevant and informative economic metrics on the herders’ disparity.
On Page 4, in 2.2, the length of subsidy payment and results were not provided.
What was your sampling method? Explain.
Provide info on details of two rounds of face-face data collection via questionnaires to farmers and herders for their responses. Did you review the first round responses from farmers and herders before conducting the second round? Explain the procedure.
How did you check whether you received an adequate number of responses and the data was sufficient for your research or objectives?
What were your rationales to choose Demographical variables?
In conclusion, how did you reach 47.89% satisfaction and 69.57% dissatisfaction due to grassland ecosystem compensation? Explain.
Conclusion missed effectiveness of implementation whether the scheme effectively achieves its intended goals, such as grassland restoration, improved livestock management, and poverty alleviation.
It's worth to be noted as limitation of study that the satisfaction of herders with such schemes could vary across regions in China due to differences in grassland ecosystems, socioeconomic conditions, and local implementation practices. Regular monitoring, evaluation, and adjustments to the scheme based on herders' feedback are essential to ensure their satisfaction and the scheme's overall success.

Round 2
Reviewer 1 Report
Comments and Suggestions for Authors
I would like to thank the authors for their thorough responses and recommend the acceptance of this paper. Congratulations!
Author Response
Dear Sir, Thank you once again for your rigorous and important comments which helped in the advancement of this manuscript.
Reviewer 2 Report
Comments and Suggestions for Authors
Good job, however merge with Review of Literature with Introduction. Do not present it as a separate section.
Author Response
Dear Sir, Thank you once again for your rigorous and important comments which helped in the advancement of this manuscript. In addition, I have separated the literature review from the introduction.
Reviewer 3 Report
Comments and Suggestions for Authors
It is better than before, suggested Accept after minor revision with proofreading.
Comments on the Quality of English LanguageIt is better than before, suggested Accept after minor revision with proofreading.
Author Response
Dear Sir, Thank you once again for your rigorous and important comments which helped in the advancement of this manuscript. In addition, I have proofread the English grammar in the manuscript and highlighted it in red.